# Attending physicians' annual service volume and use of virtual end-of-life care: A population-based cohort study in Ontario, Canada

**Rebecca Rodin[1,2], Thérèse A. Stukel[3,4], Hannah Chung[3], Chaim M. Bell[2,5,6], Allan S. Detsky[2,5,6], Sarina Isenberg[7,8,9], Kieran L. Quinn [ORCID][2,3,4,5,6] ***

1 Department of Geriatrics and Palliative Medicine, Icahn School of Medicine at Mount Sinai, New York, NY, United States of America, 2 Department of Medicine, University of Toronto, Toronto, ON, Canada, 3 ICES, Toronto and Ottawa, ON, Canada, 4 Temmy Latner Centre for Palliative Care, Toronto, ON, Canada, 5 Department of Medicine, Sinai Health System and University Health Network, Toronto, ON, Canada, 6 Institute of Health Policy, Management and Evaluation, University of Toronto, Toronto, ON, Canada, 7 Division of Palliative Care, Dept of Medicine, University of Ottawa, Ottawa, ON, Canada, 8 Bruyere Research Institute, Ottawa, ON, Canada, 9 Department of Family and Community Medicine, University of Toronto, Toronto, ON, Canada

* kieran.quinn@sinaihealth.ca

**Data Availability Statement:** The dataset from this study is held securely in coded form at ICES. While data sharing agreements prohibit ICES from

## Abstract

### Importance

Physicians and their practice behaviors influence access to healthcare and may represent potentially modifiable targets for practice-changing interventions. Use of virtual care at the end-of-life significantly increased during the COVID-19 pandemic, but its association with physician practice behaviors, (e.g., annual service volume) is unknown.

### Objective

Measure the association of physicians' annual service volume with their use of virtual end-of-life care (EOLC) and the magnitude of physician-attributable variation in its use, before and during the pandemic.

### Design, setting and participants

Population-based cohort study using administrative data of all physicians in Ontario, Canada who cared for adults in the last 90 days of life between 01/25/2018-12/31/2021. Multivariable modified Poisson regression models measured the association between attending physicians' use of virtual EOLC and their annual service volume. We calculated the variance partition coefficients for each regression and stratified by time period before and during the pandemic.

### Exposure

Annual service volume of a person's attending physician in the preceding year.

making the dataset publicly available, access may be granted to those who meet pre-specified criteria for confidential access, available at www.ices.on.ca/DAS. The full dataset creation plan and underlying analytic code are available from the authors upon request, understanding that the computer programs may rely upon coding templates or macros that are unique to ICES and are therefore either inaccessible or may require modification. ICES restricts access to information within ICES by role. Information transferred to ICES is received by an "ICES Data Covenantor," a role assigned to a limited number of ICES staff authorized to handle direct personal identifiers such as name and personal health number. The ICES Data Covenantor removes the direct personal identifiers and replaces them with a confidential ICES identifier or code to enable data linkage. The extent of access to "coded" information is then subject to access levels and permissions, which are based on need. ICES analytic staff, who create project datasets, require and therefore have access to ICES' data holdings. Others on the ICES project team are permitted to access and use project datasets only, subject to their assigned level of access. Project members who are not ICES affiliated - called "ICES collaborating researchers" – are permitted to receive information that has been summarized at a group level. ICES administers access to information on a project-by-project basis. Permission to access and use information for an ICES project is subject to a privacy impact assessment and approval by ICES' Privacy and Legal Office. Individuals who are external to ICES may access certain information for their own research subject to research ethics board approval, and are permitted to access de-identified information only.

**Funding:** This study received funding from the Canadian Institutes of Health Research (CIHR PNN- 177923), the Mount Sinai Hospital-University Health Network AMO Innovation Fund, and Health Canada, Health Care Policy and Strategies Program. This study was also supported by ICES, which is funded by an annual grant from the Ontario Ministry of Health and the Ministry of Long-Term Care. The funders had no role in study design, data collection and analysis, decision to publish, or preparation of the manuscript.

**Competing interests:** The authors have declared that no competing interests exist.

## Main outcomes and measures

Delivery of $\geq 1$ virtual EOLC visit by a person's attending physician and the proportion of variation in its use attributable to physicians.

## Results

Among the 35,825 unique attending physicians caring for 315,494 adults, use of virtual EOLC was associated with receiving care from a high compared to low service volume attending physician; the magnitude of this association diminished during the pandemic (adjusted RR 1.25 [95% CI 1.14, 1.37] pre-pandemic;1.10 (95% CI 1.08, 1.12) during the pandemic). Physicians accounted for 36% of the variation in virtual EOLC use pre-pandemic and 12% of this variation during the pandemic.

## Conclusions and relevance

Physicians' annual service volume was associated with use of virtual EOLC and physicians accounted for a substantial proportion of the variation in its use. Physicians may be appropriate and potentially modifiable targets for interventions to modulate use of EOLC delivery.

## Introduction

Physician practice behavior is an important determinant of health outcomes and patients' access to clinical care [1–6]. Such behavior may be influenced by physician characteristics, training, practice guidelines, available resources, and financial incentives [7–9]. During the COVID-19 pandemic, virtual care rapidly expanded in Ontario via introduction of new physician fee codes on March 14, 2020, which reimbursed physician delivery of video- and telephone-based virtual care, including end-of-life care (EOLC). Virtual care has the potential to expand healthcare access, improve convenience and satisfaction with care, and reduce costs through improved clinical efficiency (e.g., reduced visit length, fewer no-shows, increased on-time appointments) [10–16]. Virtual visits may be uniquely beneficial for patients near the end of life, for whom mobility outside the home and access to in-person home visiting physicians may be limited [17]. However, the optimal balance of virtual and in-person care remains to be determined.

Health systems may seek interventions to modulate use of virtual EOLC to better achieve such balance. Doing so requires measures that are easily applied and correlated with the use of virtual EOLC. While some patient-level factors, such as age [18], socioeconomic status [19], and frailty [20], may contribute to virtual EOLC use, access to this care remains largely dependent on physician providers. Physicians' use of virtual EOLC is potentially modifiable; their practice behaviors or other characteristics (e.g., age, sex, years in training, specialty) may be effective screening measures that predict which physicians are more or less likely to use virtual EOLC and therefore the most appropriate targets for intervention. At present, little is known about the extent to which such physician-level factors influence the use of this innovative EOLC delivery model.

Prior research demonstrated that physician practice behaviors, such as their annual referral rate to palliative care, was associated with their patients' subsequent use of palliative care and dying at home [2]. The Diffusion of Innovation Theory holds that some people are more apt to adopt a new technology than others (e.g., 'early adopters' versus 'laggards'), in part, because of

the former's greater need for such change [21]. Physicians with high service volume may be more likely to utilize a novel clinical delivery model, such as virtual EOLC, because of an expectation that it would improve clinical efficiency, allowing physicians to see a greater number of their patients each day [10–13]. Virtual EOLC has the potential to improve health outcomes, expand the pool of palliative care providers, and increase access to care where and when patients need it [14–16]. Conversely, overly broad use may result in unnecessary care delivery, thereby increasing overall healthcare costs [15, 22]. As a potential surrogate measure of virtual EOLC utilization, physicians' annual service volume (i.e., the number of patient visits per year) may serve as a feasible measure to identify physicians who should be targeted for interventions to modulate their virtual EOLC use. To date, no studies have measured the association of a physician's annual service volume with their use of virtual EOLC.

The objective of this study was to examine whether attending physicians' annual service volume is associated with the use of virtual EOLC. We measured the magnitude of that variation, how it changed before and during the pandemic, and quantified the variation in use of virtual EOLC attributable to physicians. We hypothesized that physicians with high annual service volumes would be more likely to delivery virtual EOLC to their patients than those with low annual service volumes.

## Methods

### Study design and setting

We used health administrative data to conduct a population-based cohort study of physicians in Ontario, Canada who provided care to adults in the last 90 days of life. These datasets were linked using unique encoded identifiers and analyzed at ICES (formerly the Institute for Clinical and Evaluative Sciences). ICES houses a vast and secure array of large, linkable and coded health-related databases including administrative and demographic datasets, population-based surveys, disease registries and electronic medical records, as well as a growing number of other non-health administrative data. The description of linked ICES datasets has been provided in S1 Table.

The Mount Sinai Research Ethics Board granted a waiver of consent for this study. ICES is a prescribed entity under Ontario's Personal Health Information Protection Act (PHIPA). Section 45 of PHIPA authorizes ICES to collect personal health information, without consent, for the purpose of analysis or compiling statistical information with respect to the management of, evaluation or monitoring of the allocation of resources to or planning for all or part of the health system. Projects that use data collected by ICES under section 45 of PHIPA, and use no other data, are exempt from REB review. The use of the data in this project is authorized under section 45 and approved by ICES' Privacy and Legal Office.

Ontario is the most populous province, with nearly 15 million residents, and provides universal access to hospital care and medically necessary physician services to all residents. These data were linked using encoded identifiers and analyzed at ICES, which have been used in multiple prior studies evaluating EOLC delivery in Ontario [2, 23–26].

### Study participants

Our study cohort included all attending physicians in Ontario who were most responsible for providing care to adults (≥18 years old) in the last 90 days of life who died from any cause over a 3-year period between January 25, 2018, and December 31, 2021. An attending physician was defined as the physician who provided the greatest number of outpatient visits for that person in the year prior to their index date, or who rostered that person as part of a primary care team. Each person had one unique attending physician.

The observation period was the last 90 days of life, beginning at an index date 90 days before death for each person. To partially disentangle the effects of the pandemic from those associated with physician service volume, we studied the periods before and during the pandemic–specifically, before and after March 14, 2020, when new specialized fee codes were introduced to enable virtual care during the pandemic. Physicians were grouped according to whether the care they provided to people in the last 90 days of life occurred before or after the introduction of specialized virtual fee codes on March 14, 2020.

We excluded people who were hospitalized for the entire observation period and those who resided in a nursing home as of the index date, as these individuals may have had limited or no access to virtual care. We also excluded individuals who had no outpatient visits in the last year of life and had not been rostered by a family physician in the last two years of life, as these individuals may be more likely to have died due to sudden, traumatic, or accidental causes, rather than due to a serious illness requiring physician care near the end of life.

## Study measures and outcomes

All variables were obtained using records from the ICES database (S1 Table). We measured physician-level characteristics, including age, sex, graduation from a Canadian versus foreign medical school, specialty, status as a palliative care specialist, years in practice, rural practice setting, number of patients and visits in the preceding year, volume of virtual and non-virtual service fee claims for the assigned patient between index date and death, volume of total physician service fee claims in the year prior to the assigned index date, and number of end-of-life visits provided in the year prior to index date. Status as a palliative care specialist was determined using a validated method with a sensitivity of 76.0% and specificity of 97.8% [27].

We also measured the characteristics of their assigned patients in terms of demographic and clinical variables, including age, sex, neighborhood income quintile, location of residence (rural and home/community service networks known as LHINs), surname-based ethnicity [28], comorbidities and chronic conditions [29], Hospital Frailty Index scores [30], recent healthcare use (receipt of palliative care before the last 90 days of life, number of medications, emergency department visits, hospitalizations, and receipt of homecare services in past year) and year of death. The Hospital Frailty Index score is a comprehensive measure of a person's comorbidity that reflects global illness severity; this score predicts greater risk of adverse outcomes, including hospitalization and 30-day mortality [30].

The main exposure was the annual service volume of a person's attending physician in the year prior to their index date. The annual service volume is defined as the number of patient visits provided by a physician each year. This was measured for each physician using the number of outpatient visits the attending physician provided to all patients in the year prior to their assigned patient's index date. We then categorized physicians into percentiles based on their relative annual service volume ("low" [bottom 25%], "high" [top 25%], and "average" [volumes within 25–75%]). We focused on comparing physicians with high vs. low annual service volume because these groups were expected to have the greatest relative difference in outcomes and efforts to modify physician behaviors, such as audit and feedback, are most effective when targeted to physicians with practice patterns at the extremes [31].

The primary outcome was the use of virtual EOLC, defined as ≥1 virtual care visit by an attending physician to their patient in the last 90 days of life. Prior to the pandemic, the only virtual care reimbursed by the universal healthcare plan in Ontario, Canada was for a maximum of two telephone encounters per week, or for specific video-based encounters that required patients to travel to an authorized center. During the pandemic, new specialized fee codes were introduced that reimbursed all video- and telephone-based encounters occurring

anywhere within the province, including from a patient's home. We also measured the proportion of variation in use of virtual EOLC attributable to physicians. The secondary outcomes were: 1) the number of unique visits of any kind provided by an attending physician to their assigned patient during the last 90 days of life; 2) the number of unique virtual visits provided by an attending physician to their assigned patient during the last 90 days of life; and 3) the proportion of virtual visits to total unique visits provided by the attending physician to their assigned patient from index date to death.

## Data analysis

Modified Poisson regression was used to measure the association (relative risk) between the use of end-of-life virtual care by attending physicians for their patients and the annual service volume of those physicians, accounting for clustering of people within the same physician. We stratified these analyses by time period before and during the pandemic in order account for the widespread changes in healthcare utilization that occurred during the pandemic. Covariates included in the analytic models were chosen as potential confounders based on the clinical and research expertise of our team, including those that were imbalanced based on a measured standardized differences (SD) assessed at index date of >0.1 [32]. For example, the inclusion of physician age, sex, and specialty as covariates was based on studies showing that younger, female physicians in primary practice care for a fewer number of patients than physicians who are older, male, and specialized; and so these variables may confound measures of annual service volume [33, 34]. For each regression model, we calculated the variance partition coefficient (VPC), which measures the proportion of the variation in use of virtual EOLC that is attributable to physicians. All analyses were performed using SAS version 9.4 (SAS Institute, Cary, North Carolina).

## Results

### Characteristics of study participants

We included 35,825 unique attending physicians who were paired with 315,494 adults who died during the study period. Prior to the pandemic, there were 18,335 attending physicians, of whom 7,434 (41%) were low, 8,099 (44%) average, and 2,802 (15%) high volume providers. During the pandemic, there were 17,490 attending physicians, of whom 7,876 (45%) were low, 7,221 (41%) average, and 2,393 (14%) high volume providers (Table 1; S2 Table). Among the 315,494 patients in the study, there were 77,826 (25%) people paired with low annual service volume physicians, 158,068 (50%) people paired with average annual service volume physicians, and 79,600 (25%) people paired with high annual service volume physicians (Fig 1).

Within each period, a greater proportion of high compared to low annual volume physicians were male, trained in family medicine, practiced in urban areas, graduated from an international medical school, and, by definition, conducted more patient visits per year (Table 1; S2 Table). The magnitude of these differences were not different before and during the pandemic. The characteristics of the people cared for by each attending physician in each period is shown in S3 Table.

### Changes in virtual EOLC by physician annual service volume

Before the pandemic, 4.3% of people cared for by high volume attending physicians and 4.2% of people cared for by low volume attending physicians received at least 1 virtual EOLC visit. During the pandemic, these proportions rose to 58% and 52%, respectively (Table 2). After adjusting for physician- and patient-level characteristics, use of virtual EOLC was associated with receiving care from a high compared to low volume attending physician, though the

**Table 1. Baseline physician characteristics according to annual service volume before and during the pandemic.**

| Baseline Physician Characteristic | Before the Pandemic | | | During the Pandemic | | |
|---|---|---|---|---|---|---|
| | Annual Physician Practice Volume* | | | Annual Physician Practice Volume* | | |
| | Low (N = 7,434) | High (N = 2,802) | Standardized Difference | Low (N = 7,876) | High (N = 2,393) | Standardized Difference |
| Age (y), mean (SD) | 50.1 (13.3) | 51.8 (10.6) | 0.14 | 51.0 (13.0) | 52.3 (10.5) | 0.11 |
| Female sex, n (%) | 3,634 (48.9) | 629 (22.4) | 0.57 | 3,746 (47.6) | 566 (23.7) | 0.52 |
| Rural, n (%) | 502 (6.8) | 55 (2.0) | 0.24 | 600 (7.6) | 33 (1.4) | 0.3 |
| Canadian medical graduate, n (%) | 4,857 (65.3) | 1,396 (49.8) | 0.32 | 4,935 (62.7) | 1,127 (47.1) | 0.32 |
| Years in practice, median (IQR) | 22 (12–35) | 27 (18–35) | 0.22 | 23 (12–36) | 27 (19–35) | 0.21 |
| Practice specialty, n (%) | | | | | | |
| Family Medicine | 4,008 (53.9) | 1,941 (69.3) | 0.32 | 4,458 (56.6) | 1,643 (68.7) | 0.25 |
| Non-Cancer Specialist | 2,422 (32.6) | 670 (23.9) | 0.19 | 2,407 (30.6) | 568 (23.7) | 0.15 |
| Medical Oncologist | 207 (2.8) | 39 (1.4) | 0.1 | 173 (2.2) | 53 (2.2) | 0 |
| Surgical Oncologist | 745 (10.0) | 151 (5.4) | 0.17 | 834 (10.6) | 129 (5.4) | 0.19 |
| Status as a palliative care specialist, n (%) | 331 (4.5) | 66 (2.4) | 0.12 | 352 (4.5) | 97 (4.1) | 0.02 |
| Number of patients per physician | | | | | | |
| Median (IQR) | 3 (1–7) | 11 (6–21) | 1.17 | 3 (1–6) | 10 (5–19) | 1.2 |
| Number of visits in year prior to index | 2,221 (1,499–2,826) | 9,678 (8,288–12,244) | 3.16 | 2,187 (1,450–2,784) | 9,633 (8,255–12,095) | 3.06 |
| Median (IQR) | | | | | | |
| Number of end-of-life visits in year prior to index** | | | | | | |
| Median (IQR) | 0 (0–0) | 0 (0–1) | 0.08 | 2 (0–7) | 10 (3–22) | 0.81 |

SD indicates standard deviation; IQR, interquartile range; MRP, most responsible physician

*See S2 Table for complete table showing physician demographics according to low, average, and high annual service volume before and during the pandemic

**An end-of-life visit is defined by the presence of any of the following billings: 1) Home-based palliative care, 2) Virtual palliative care, 3) Any virtual visit billed with a home-based palliative care fee code in the last year of life, 4) Any virtual visit within the last 90 days of life

magnitude of this association diminished during the pandemic (adjusted RR 1.25 [95% CI 1.14, 1.37] pre-pandemic and 1.10 [95% CI 1.08, 1.12] during the pandemic; Table 2). This was also observed for people with high compared to average volume physicians (adjusted RR 1.20 [95% CI 1.11, 1.30] pre-pandemic and 1.02 [95% CI 1.00, 1.03] during the pandemic; Table 2). Overall, the increase in EOLC during the pandemic included a nearly 20-fold increase in the proportion of EOLC visits that occurred virtually (Table 3). Measurement of the VPC showed that physicians accounted for 36% of the variation in use of virtual EOLC in the pre-pandemic period and 12% of this variation during the pandemic (Fig 2).

### Changes in any EOLC by physician annual service volume

The delivery of any EOLC visits (in-person or virtual) by an attending physician increased in a similar manner to virtual EOLC according to annual physician service volume (Table 3). The delivery of unique EOLC visits of any kind increased during the pandemic, with high compared to low volume physicians delivering more EOLC visits to their assigned patients (2.2 vs. 1.92 visits pre-pandemic; 3.08 vs. 2.18 visits during the pandemic) (Table 3).

## Discussion

This population-based study of 35,825 attending physicians most responsible for the care of 315,494 adults at the end of life measured the association between annual physician service

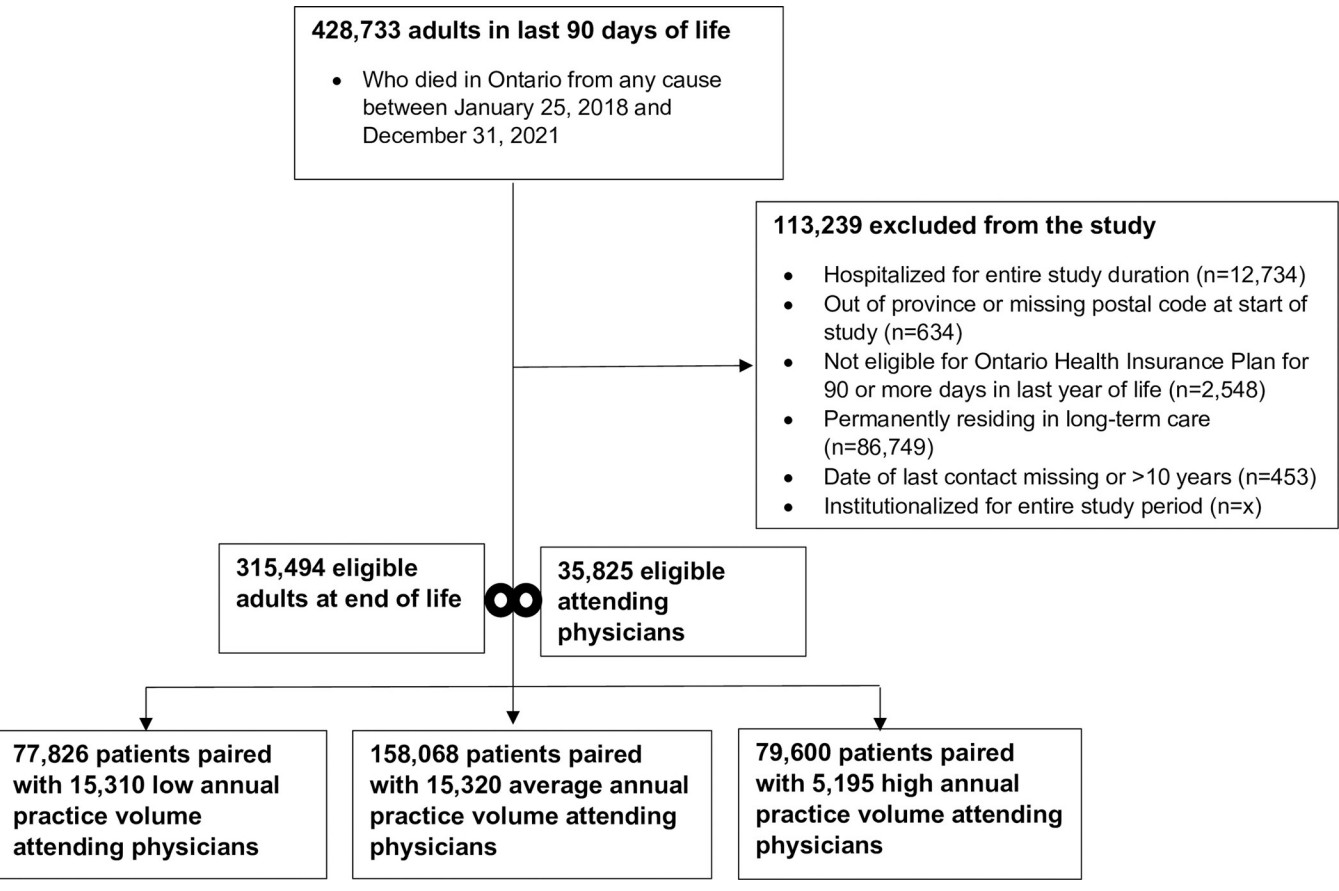

**Fig 1. Creation of the study cohort.**

volume and the use of virtual EOLC. We found that attending physicians with a high annual service volume were significantly more likely to use virtual EOLC compared to attending physicians with a low annual service volume. The magnitude of this association was attenuated during the pandemic. Further analysis demonstrated that attending physicians accounted for a significant proportion of the variation in the use of virtual EOLC. Taken together, these findings suggest that physician practice behavior (specifically, their annual service volume) and attending physicians themselves play a significant role in the use of virtual EOLC.

The present findings are consistent with prior work showing the impact of physician practice behaviors on clinical care, such as referral to palliative care [1, 4, 23, 35–37]. While patients accounted for most of the variation in receipt of virtual EOLC across both time periods, a substantial proportion remained attributable to physicians. Therefore, physicians may present modifiable targets for interventions to effect practice change in the use of virtual EOLC, which may include targeted audit and feedback, education, financial, and policy initiatives [31, 38].

The findings in this study suggest that annual service volume may be an easy and feasible method for identifying physicians most likely to respond to such targeted efforts to either increase or decrease their use of virtual EOL. At present, little is known about the quality, efficacy, and effectiveness of virtual EOLC, its impact on healthcare outcomes, and the optimal balance of in-person and virtual visits. Most research on virtual care was conducted before the onset of the COVID-19 pandemic, when virtual care was sparingly utilized [16, 39–42]. Recently published research in this domain has yielded inconsistent results, with some studies

**Table 2. Association between attending physician service volume and use of virtual EOLC to assigned patients according to time period before and during the pandemic.**

| Annual Physician Service Volume | People Receiving ≥1 virtual EOLC Visit by Attending Physician, n (%) | Unadjusted Relative Risk (95% C.I.) | Adjusted* Relative Risk (95% C.I.) |
|---|---|---|---|
| Before the Pandemic | | - | - |
| High | 1,954 (4.3) | | |
| Average | 3,596 (4.0) | | |
| Low | 1,695 (4.2) | - | - |
| During the Pandemic | | | |
| High | 19,823 (57.7) | | |
| Average | 38,144 (56.5) | - | - |
| Low | 19,377 (52.0) | | |
| High vs. Low | | | |
| Before- | | 1.03 (0.97, 1.10) | 1.25 (1.14, 1.37) |
| During- | | 1.11 (1.10, 1.13) | 1.10 (1.08, 1.12) |
| Average vs. Low | | | |
| Before- | | 0.95 (0.90, 1.00) | 1.03 (0.96, 1.10) |
| During- | | 1.09 (1.08, 1.10) | 1.07 (1.06, 1.09) |
| High vs. Average | | | |
| Before- | | 1.09 (1.03, 1.15) | 1.20 (1.11, 1.30) |
| During- | | 1.02 (1.01, 1.03) | 1.02 (1.00, 1.03) |

* Model adjusted for the following provider-level characteristics: sex, age, practice location (urban vs. rural), practice type as specialist vs. family physician/general practitioner, status as a palliative care billing physician, and number of years in practice; and patient-level characteristics: age, sex, neighborhood income quintile, surname-based ethnicity, rural residence, local health integration network (LHIN), chronic conditions, hospital frailty risk score, homecare in the 2 years prior to index, number of unique medications, acute healthcare use and use of palliative care in the year prior to index

showing that an overreliance on virtual care is associated with increased healthcare utilization, including repeat virtual or in-person visits, hospitalizations, and emergency department visits [22, 43–45]. Other studies found no such association or mixed results, in addition to high patient satisfaction with virtual care [46–50], and public polling indicates that people want continued access to virtual care, particularly for chronic and non-urgent issues [51]. Ongoing efforts to establish the optimal balance of virtual and in-person EOLC can inform physician-targeted interventions to modulate its use, either by increasing or decreasing utilization of such care.

Strengths of this study include its use of a large, population-based sample that minimizes selection bias and the inclusion of a diverse sample of physicians who provided care near the

**Table 3. Delivery of virtual and non-virtual care by a person's attending physician in the last 90 days of life according to annual physician service volume, before and during the pandemic.**

| | Before the Pandemic* | | During the Pandemic* | |
|---|---|---|---|---|
| | Low Annual Physician Volume | High Annual Physician Volume | Low Annual Physician Volume | High Annual Physician Volume |
| Unique virtual EOLC visits delivered to patients by attending physician, mean (SD) | 0.06 (0.3) | 0.15 (0.6) | 1.14 (1.7) | 1.83 (1.7) |
| Unique EOLC visits (in person or virtual) delivered to patients by attending physician, mean (SD) | 1.92 (1.8) | 2.22 (1.5) | 2.18 (2.2) | 3.08 (2.2) |
| Proportion of virtual visits to total unique EOLC visits, % | 3.1 | 6.8 | 52.3 | 59.4 |

*See S4 Table for complete table showing results according to low, average, and high annual service volume before and during the pandemic

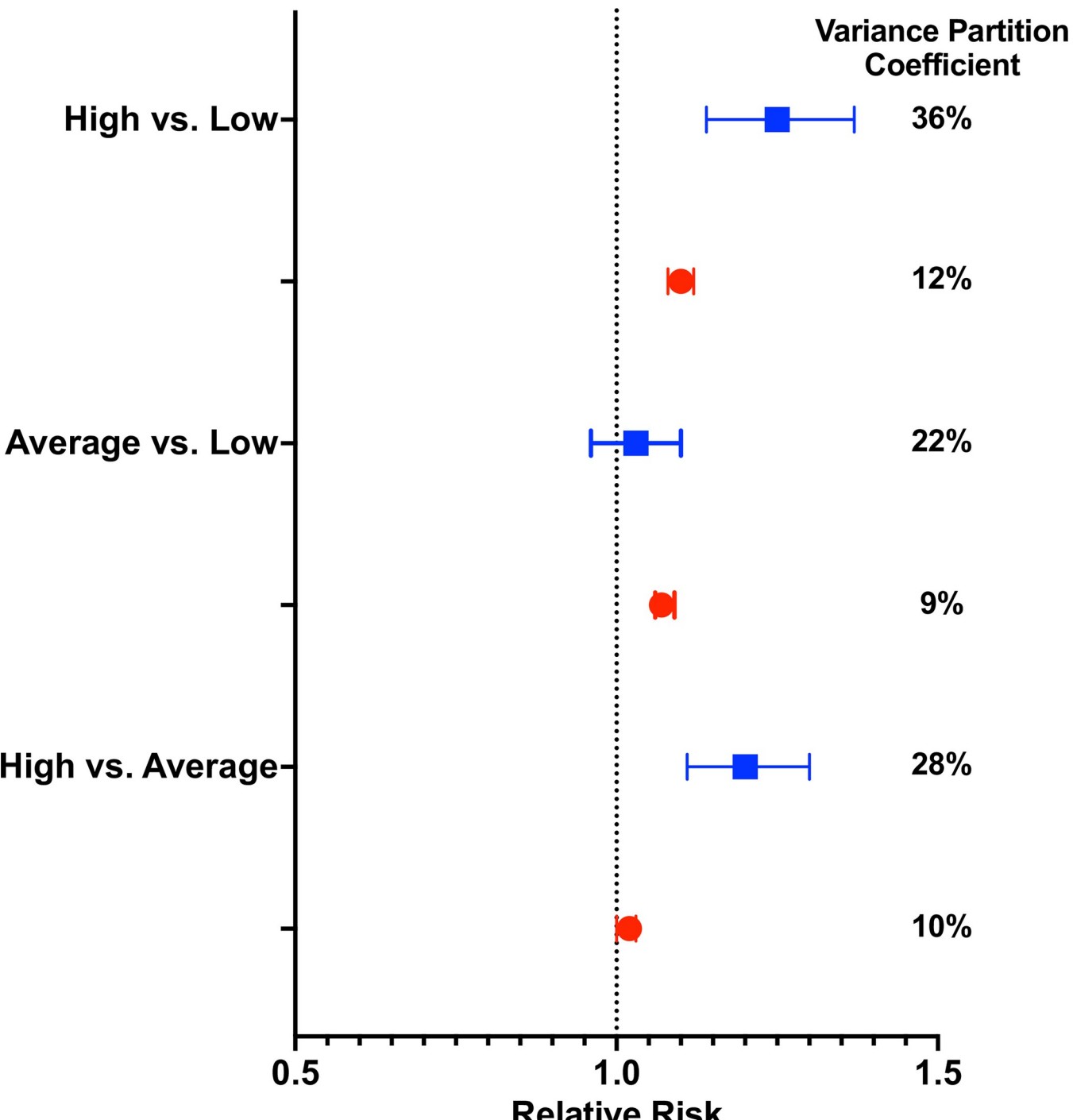

**Fig 2. Primary analysis.** Forest plot of the adjusted relative risk of the association between attending physicians' annual service volume with the use of virtual end-of-life care for the pre-March 14, 2020 (blue squares) and post-March 14, 2020 (red circles) groups. The variance partition coefficient (VPC) estimated the magnitude of physician-attributable variation in the use of virtual end-of-life care.

end of life. The massive shift to virtual care delivery during the pandemic due to restrictions on in-person care allowed for the study of virtual EOLC utilization across an entire health system. Further, the relative stability of our sample characteristics before and during the

pandemic minimizes the likelihood that our findings are confounded by changes in the types of physicians and patients utilizing virtual EOLC during the pandemic.

Our study also has limitations. First, we did not measure individual patient preferences and their ability to engage in virtual EOLC. Many patients still prefer virtual care, or desire it as a supplement to in-person care, in multiple circumstances including near end-of-life [17, 52], although others may be less able or willing to access virtual EOLC [53–55]. Second, the observational design of our study limits our understanding of the causal mechanisms related to our primary findings, which could be partly related to shifts in healthcare delivery during the pandemic. We explored this potential confounding bias by measuring the magnitude of the effects in the pre-pandemic and pandemic periods. The attenuated magnitude of our findings during the pandemic suggests that changes in healthcare delivery during the pandemic may have also had an important effect on virtual EOLC utilization. Third, we lacked information on the nature and quality of care delivered for each virtual EOLC visit, such as whether it focused primarily on palliative and supportive needs or acute or chronic medical management. Specialized fee codes for palliative-specific virtual care were not introduced in Ontario until January 2021, after data collection for this study was completed. Further research is needed to examine the impact of these palliative-specific virtual billing codes on overall end-of-life-care delivery. Fourth, selection bias may have arisen due to the inclusion of physicians who treat people near the end of life, but not of physicians who only treat people earlier in the course of illness. While these characteristics may affect the likelihood to use virtual care technology, this potential bias was mitigated by including a diverse sample of physicians across the health system. Fifth, the study findings may vary according to physician specialty, which we did not specifically examine, and may affect its generalizability, despite adjustment for physicians' status as either a primary care physician, palliative or other specialist in our analytic models. Sixth, this study focused exclusively on end-of-life care in Ontario, Canada, so generalizability to other domains of care or to other health systems is not yet known.

Future research is needed to develop and evaluate physician-targeted interventions to effect practice change among physicians to either expand or restrict their utilization of virtual EOLC depending on the desired outcome. Policy-makers and health systems developing such interventions may utilize measures of physicians' annual service volume to identify the physicians most suitable for these interventions, as such measures are easily applied and correlated with virtual EOLC utilization.

## Conclusions

Physician practice behavior, as reflected by attending physicians' annual service volumes, is associated with the use of virtual EOLC and physicians account for a substantial proportion of the variation in its use. Physicians may therefore be an appropriate and potentially modifiable target for interventions to help modulate and standardize access to this new model of EOLC delivery.

## Supporting information

**S1 Checklist. STROBE statement- checklist of items that should be included in reports of cohort studies.**
(PDF)

**S1 Table. Description of linked ICES datasets.**
(DOCX)

**S2 Table. Baseline physician characteristics according to annual service volume before and during the pandemic.**
(DOCX)

**S3 Table. Baseline characteristics of all community-dwelling adults in the last 90 days of life who died in Ontario according to their attending physician's annual service volume before and during the pandemic.**
(DOCX)

**S4 Table. Delivery of virtual and non-virtual care by a person's attending physician in the last 90 days of life according to annual physician service volume, before and during the pandemic.**
(DOCX)

## Acknowledgments

The authors would like to express their gratitude to our patient and caregiver partner who assisted in the development of the research questions and interpretation of the main findings. They politely declined the offer to include them as co-authors in recognition of their valuable contributions. We thank IQVIA Solutions Canada Inc. for use of their Drug Information File.

## Author Contributions

**Formal analysis:** Hannah Chung.

**Supervision:** Kieran L. Quinn.

**Writing – original draft:** Rebecca Rodin.

**Writing – review & editing:** Thérèse A. Stukel, Chaim M. Bell, Allan S. Detsky, Sarina Isenberg.

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
