## [Decision Letter · Decision Letter 0]

19 Sep 2023

PONE-D-23-24605Attending Physicians’ Annual Service Volume and Use of Virtual End-of-Life Care: A Population-Based Cohort StudyPLOS ONE

Dear Dr. Quinn,

Thank you for submitting your manuscript to PLOS ONE. After careful consideration, we feel that it has merit but does not fully meet PLOS ONE’s publication criteria as it currently stands. Therefore, we invite you to submit a revised version of the manuscript that addresses the points raised during the review process.

We look forward to receiving your revised manuscript.

Kind regards,

Masoud Behzadifar

Academic Editor

PLOS ONE

Journal Requirements:

2. You indicated that ethical approval was not necessary for your study. We understand that the framework for ethical oversight requirements for studies of this type may differ depending on the setting and we would appreciate some further clarification regarding your research. Could you please provide further details on why your study is exempt from the need for approval and confirmation from your institutional review board or research ethics committee (e.g., in the form of a letter or email correspondence) that ethics review was not necessary for this study? Please include a copy of the correspondence as an ""Other"" file.

4. Please note that funding information should not appear in any section or other areas of your manuscript. We will only publish funding information present in the Funding Statement section of the online submission form. Please remove any funding-related text from the manuscript.

6. Thank you for stating the following financial disclosure: 

   "NO"

7. In your Data Availability statement, you have not specified where the minimal data set underlying the results described in your manuscript can be found. PLOS defines a study's minimal data set as the underlying data used to reach the conclusions drawn in the manuscript and any additional data required to replicate the reported study findings in their entirety. All PLOS journals require that the minimal data set be made fully available. For more information about our data policy, please see http://journals.plos.org/plosone/s/data-availability.

8. We note that you have stated that you will provide repository information for your data at acceptance. Should your manuscript be accepted for publication, we will hold it until you provide the relevant accession numbers or DOIs necessary to access your data. If you wish to make changes to your Data Availability statement, please describe these changes in your cover letter and we will update your Data Availability statement to reflect the information you provide.

9. Please include your full ethics statement in the ‘Methods’ section of your manuscript file. In your statement, please include the full name of the IRB or ethics committee who approved or waived your study, as well as whether or not you obtained informed written or verbal consent. If consent was waived for your study, please include this information in your statement as well. 

Reviewers' comments:

Reviewer's Responses to Questions

**Comments to the Author**

1. Is the manuscript technically sound, and do the data support the conclusions?

Reviewer #1: Partly

Reviewer #2: Partly

2. Has the statistical analysis been performed appropriately and rigorously? 

Reviewer #1: Yes

Reviewer #2: I Don't Know

3. Have the authors made all data underlying the findings in their manuscript fully available?

Reviewer #1: No

Reviewer #2: No

4. Is the manuscript presented in an intelligible fashion and written in standard English?

Reviewer #1: No

Reviewer #2: Yes

5. Review Comments to the Author

Reviewer #1: 1- the introduction effectively sets the context for the study, however, it would be helpful to include more background information such as benefits of telemedicine and its economic impact on reducing costs in providing services.

2-in method section it would be beneficial to explain how attending physicians services volume was defined and measured. Additionally, more information about the confounding variables such as age, sex , specialist types would enhance the quality of the study.

3-it would be worthwhile to include more studies in discussion section.

4-researchers must have an ethical responsibility to report their finding accurately and responsibly.

Reviewer #2: • In title, please specify the location of this study.

• I think you should define and explain the "annual service volume".

• The first paragraph of introduction is too long. Please concise it for better readability.

• Has the protocol of this study been previously registered and indexed somewhere that is publicly available?

• Please use relevant reporting guideline throughout the study (https://www.equator-network.org/wp-content/uploads/2015/10/STROBE_checklist_v4_cohort.pdf ), and upload a copy of completed checklist as supplementary file.

• In discussion, you should discuss the potential biases and effect of them on your findings.

6. PLOS authors have the option to publish the peer review history of their article (what does this mean?). If published, this will include your full peer review and any attached files.

Reviewer #1: No

Reviewer #2: No

---

## [Author Response · Author response to Decision Letter 0]

12 Jan 2024

Dr. Masoud Behzadifar

Academic Editor

PLOS ONE 

PONE-D-23-24605 “Attending Physicians’ Annual Service Volume and Use of Virtual End-of-Life Care: A Population-Based Cohort Study in Ontario, Canada.” 

Thank you for providing us the opportunity to resubmit our work to PLOS One. We are pleased that the editors and reviewers found our study to be important, valuable, relevant, and of considerable interest. Their helpful suggestions and comments have helped us to improve the paper. We have addressed their concerns in detail below.

The following is a summary of the extensive changes we made to the manuscript in response to yours and the reviewer’s suggestions:

1) Added more fulsome discussion with additional citations to place work in context of current understanding and knowledge about telemedicine. 

2) Provided additional background information and citations on the potential benefits of telemedicine.

3) Added details on covariates and a supplementary table providing detail of datasets used to measure them.

4) Further discussed potential sources of bias.

5) Clarified methodological details on the definition of a physician’s annual service volume and its measurement. 

6) Modified the title to include the location of study.

7) Provided detail on research ethics exemption and designated status of ICES as a prescribed entity in Ontario.

8) Updated formatting of the manuscript to align with journal-specific requirements.

1. Journal Requirements

1.1 Please ensure that your manuscript meets PLOS ONE’s style requirements, including those for file naming. 

We have updated the manuscript as per PLOS ONE's style requirements.

1.2 Could you please provide further details on why your study is exempt from the need for approval and confirmation from your institutional review board or research ethics committee (e.g., in the form of a letter or email correspondence) that ethics review was not necessary. Please include a copy of the correspondence as an “Other” file.

The administrative datasets used in this study were linked using encoded identifiers at the person-level at ICES (formerly the Institute for Clinical and Evaluative Sciences). ICES is a prescribed entity under Ontario’s Personal Health Information Protection Act (PHIPA). Section 45 of PHIPA authorizes ICES to collect personal health information, without consent, for the purpose of analysis or compiling statistical information with respect to the management of, evaluation or monitoring of, the allocation of resources to or planning for all or part of the health system. Projects that use data collected by ICES under section 45 of PHIPA, and use no other data, are exempt from REB review. The use of the data in this project is authorized under section 45 and approved by ICES’ Privacy and Legal Office. Details about this exemption have been added to the manuscript (Methods, page 5, lines 188-195).

We have included the copy of the waiver from our institutional review board as an “Other” file with our submission.

1.3. Please provide additional details regarding participant consent [and] ensure that you have specified what type of consent you obtained. …If the need for consent was waived… please include this information.

We have uploaded the waiver letter from our institutional Research Ethics Board, updated the 'Ethics Statement' field of the submission form and added rationale in the methods section for not obtaining participant consent as follows,

“The Mount Sinai Research Ethics Board granted a waiver of consent for this study. ICES is a prescribed entity under Ontario’s Personal Health Information Protection Act (PHIPA). Section 45 of PHIPA authorizes ICES to collect personal health information, without consent, for the purpose of analysis or compiling statistical information with respect to the management of, evaluation or monitoring of the allocation of resources to or planning for all or part of the health system. Projects that use data collected by ICES under section 45 of PHIPA, and use no other data, are exempt from REB review. The use of the data in this project is authorized under section 45 and approved by ICES’ Privacy and Legal Office.” (Methods, page 5, lines 188-195)

1.4. Please remove any funding-related text from the manuscript. 

We have removed funding-related text from the manuscript.

1.5. We note that the grant information you provided in the ‘Funding Information’ and ‘Financial Disclosure’ sections do not match… Please ensure that you provide the correct grant numbers for the awards you received for your study in the ‘Funding Information’ section.

We have updated the Funding Information’ and ‘Financial Disclosure, ensured that they match, and have added grant numbers for the awards. 

1.6. Please state what role the funders took in the study. If the funders had no role, please state: "The funders had no role in study design, data collection and analysis, decision to publish, or preparation of the manuscript." If this statement is not correct you must amend it as needed. Please include this amended Role of Funder statement in your cover letter; we will change the online submission form on your behalf.

We have now added the suggested statement to the online financial disclosure form: “The funders had no role in study design, data collection and analysis, decision to publish, or preparation of the manuscript.”

1.7. In your Data Availability statement, you have not specified where the minimal data set underlying the results described in your manuscript can be found… Please upload your study’s minimal underlying data set as either Supporting Information files or to a stable, public repository and include the relevant URLs, DOIs, or accession numbers within your revised cover letter. For a list of acceptable repositories, please see http://journals.plos.org/plosone/s/data-availability#loc-recommended-repositories. Any potentially identifying patient information must be fully anonymized.

Important: If there are ethical or legal restrictions to sharing your data publicly, please explain these restrictions in detail.

We previously indicated our data availability in our manuscript as follows,

“The dataset from this study is held securely in coded form at ICES. While data sharing agreements prohibit ICES from making the dataset publicly available, access may be granted to those who meet pre-specified criteria for confidential access, available at www.ices.on.ca/DAS. The full dataset creation plan and underlying analytic code are available from the authors upon request, understanding that the computer programs may rely upon coding templates or macros that are unique to ICES and are therefore either inaccessible or may require modification.” (Data Sharing, page 16, lines 539-544)

1.8. We note that you have stated that you will provide repository information for your data at acceptance. Should your manuscript be accepted for publication, we will hold it until you provide the relevant accession numbers or DOIs necessary to access your data. If you wish to make changes to your Data Availability statement, please describe these changes in your cover letter and we will update your Data Availability statement to reflect the information you provide.

Please see our response #1.7 on data access.

1.9. Please include your full ethics statement in the ‘Methods’ section of your manuscript file. In your statement, please include the full name of the IRB or ethics committee who approved or waived your study, as well as whether or not you obtained informed written or verbal consent. If consent was waived for your study, please include this information in your statement as well. 

We added the ethics statement in the “Methods” section as follows, 

“The Mount Sinai Research Ethics Board granted a waiver of consent for this study. ICES is a prescribed entity under Ontario’s Personal Health Information Protection Act (PHIPA). Section 45 of PHIPA authorizes ICES to collect personal health information, without consent, for the purpose of analysis or compiling statistical information with respect to the management of, evaluation or monitoring of the allocation of resources to or planning for all or part of the health system. Projects that use data collected by ICES under section 45 of PHIPA, and use no other data, are exempt from REB review. The use of the data in this project is authorized under section 45 and approved by ICES’ Privacy and Legal Office.” (Methods, page 5, lines 188-195)

2. Reviewer Comments

2.1. In title, please specify the location of the study [R2]. 

The study title has been revised to include the location of the study in Ontario, Canada (page 1, line 3).

2.2. It would be helpful to include more background information [in the introduction], such as benefits of telemedicine, its economic impact on reducing costs in providing services [R1]. 

We added additional background information and references on the potential benefits of telemedicine as follows,

“Virtual care has the potential to expand healthcare access, improve convenience and satisfaction with care, and reduce costs through improved clinical efficiency (e.g., reduced visit length, fewer no-shows, increased on-time appointments). [10-16] Virtual visits may be uniquely beneficial for patients near the end of life, for whom mobility outside the home and access to in-person home visiting physicians may be limited.[17] However, the optimal balance of virtual and in-person care remains to be determined.” (Introduction, page 3, lines 112-117).

Citations:

10. Green, L.V., et al., The Impact of Telehealth on Primary Care Physician Panel Sizes: A Modeling Study. J Am Board Fam Med, 2022. 35(5): p. 1007-1014.

11. Andino, J.J., et al., The Impact of Video Visits on Measures of Clinical Efficiency and Reimbursement. Urol Pract, 2021. 8(1): p. 53-57.

12. Pennington, Z., et al., Positive impact of the pandemic: the effect of post-COVID-19 virtual visit implementation on departmental efficiency and patient satisfaction in a quaternary care center. Neurosurg Focus, 2022. 52(6): p. E10.

13. Tsaousis, K.T., et al., The concept of virtual clinics in monitoring patients with age-related macular degeneration. Acta Ophthalmol, 2016. 94(5): p. e353-5.

14. Slavin-Stewart, C., A. Phillips, and R. Horton, A Feasibility Study of Home-Based Palliative Care Telemedicine in Rural Nova Scotia. J Palliat Med, 2020. 23(4): p. 548-551.

15. Steindal, S.A., et al., Patients' Experiences of Telehealth in Palliative Home Care: Scoping Review. J Med Internet Res, 2020. 22(5): p. e16218.

16. Zheng, Y., B.A. Head, and T.J. Schapmire, A Systematic Review of Telehealth in Palliative Care: Caregiver Outcomes. Telemed J E Health, 2016. 22(4): p. 288-94.

17. Vincent, D., et al., Virtual home-based palliative care during COVID-19: A qualitative exploration of the patient, caregiver, and healthcare provider experience. Palliative Medicine, 2022. 36(9): p. 1374-1388.

2.3. The first paragraph is too long [R2]. 

We revised the first paragraph to be more concise. (Introduction, page 3, lines 110-125).

2.4. In method section it would be beneficial to explain how attending physicians services volume was defined and measured [R1]. Define and explain the “annual service volume” [R2].

We clarified additional details on the definition of a physician’s annual service volume and its measurement as follows, 

“. The annual service volume is defined as the number of patient visits provided by a physician each year. This was measured for each physician using the number of outpatient visits the attending physician provided to all patients in the year prior to their assigned patient’s index date. We then categorized physicians into percentiles based on their relative annual service volume (“low” [bottom 25%], “high” [top 25%], and “average” [volumes within 25-75%]).” (Methods, page 7, lines 258-262).

We added the following clarification to the introduction, 

“…physicians’ annual service volume (i.e., the number of patient visits per year) may serve as a feasible measure to identify physicians for targeted interventions to modulate virtual EOLC use.” (Introduction, page 4, lines 154-155)

2.5. More information about the confounding variables, such as age, sex, specialist types, would enhance the quality of the study [R1]. 

We provided additional information about confounding variables as follows,

“the inclusion of physician age, sex, and specialty as covariates was based on studies showing that younger, female physicians in primary practice care for a fewer number of patients than physicians who are older, male, and specialized; and so these variables may confound measures of annual service volume.[33, 34]” (Methods, page 8, lines 305-308)

We also added a supplementary table describing the linked datasets used in this study, including the ICES Physician Database that describes how physician characteristics were measured (S1 Table).

2.6. It would be worthwhile to include more studies in discussion section [R1].

Additional studies have been included in the discussion section (references 1, 4, 35-38, 44, 45, 48). References 1, 4, 35-37 (Discussion, page 13, line 406) are studies on the impact of physician practice behaviors, such as their referral patterns and practice volume, on health outcomes; reference 38 is a systematic review of the use of targeted audit and feedback interventions to improve physician performance (page 13, line 410); and references 44, 45, and 48 are studies on the impact of virtual care on subsequent healthcare utilization (page 14, lines 423-424).

2.5. In discussion, you should discuss the potential biases and effect of them on your findings [R2].

Further discussion of potential biases and their effect on our findings have been added to the discussion as follows,

“Second, the observational design of our study limits our understanding of the causal mechanisms related to our primary findings, which could be partly related to shifts in healthcare delivery during the pandemic. We explored this potential confounding bias by measuring the magnitude of the effects in the pre-pandemic and pandemic periods. The attenuated magnitude of our findings during the pandemic suggests that changes in healthcare delivery during the pandemic may have also had an important effect on virtual EOLC utilization. Third, we lacked information on the nature and quality of care delivered for each virtual EOLC visit, such as whether it focused primarily on palliative and supportive needs or acute or chronic medical management. Specialized fee codes for palliative-specific virtual care were not introduced in Ontario until January 2021, after data collection for this study was completed. Further research is needed to examine the impact of these palliative-specific virtual billing codes on overall end-of-life-care delivery. Fourth, selection bias may have arisen due to the inclusion of physicians who treat people near the end of life, but not of physicians who only treat people earlier in the course of illness. While these characteristics may affect the likelihood to use virtual care technology, this potential bias was mitigated by including a diverse sample of physicians across the health system. Fifth, the study findings may vary according to physician specialty, which we did not specifically examine, and may affect its generalizability, despite adjustment for physicians’ status as either a primary care physician, palliative or other specialist in our analytic models. Sixth, this study focused exclusively on end-of-life care in Ontario, Canada, so generalizability to other domains of care or to other health systems is not yet known.” (page 14-15, lines 439-506).

2.6. Report… funding accurately and responsibly [R1].

All funding-related text has been removed from the manuscript. Funding has been reported accurately and consistently in the ‘Funding Information’ section. See response #1.7 above.

2.7. Has the protocol of this study been previously registered and indexed somewhere that is publicly available? [R2].

The study protocol was not previously registered or indexed in a publicly available location. Upon request, the dataset creation plan can be provided as per ICES policy.

2.8. Please use relevant reporting guideline throughout the study (https://www.equator-network.org/wp-content/uploads/2015/10/STROBE_checklist_v4_cohort.pdf ), and upload a copy of completed checklist as supplementary file [R2].

The manuscript has been revised throughout based on the STROBE reporting guideline. A completed checklist is submitted as a supplementary file.

Thank you in advance for your consideration,

Kieran L. Quinn, MD PhD

E-mail: kieran.quinn@sinaihealth.ca

Assistant Professor, Department of Medicine, University of Toronto, Toronto, ON, Canada 

Clinician-Scientist, Sinai Health System and University Health Network, Toronto, ON, Canada

Adjunct Scientist, ICES, Toronto and Ottawa, ON, Canada 

Institute of Health Policy, Management and Evaluation, University of Toronto, Toronto, ON, Canada

---

## [Decision Letter · Decision Letter 1]

16 Feb 2024

Attending physicians’ annual service volume and use of virtual end-of-life care: a population-based cohort study in Ontario, Canada

PONE-D-23-24605R1

Dear Dr. Quinn,

We’re pleased to inform you that your manuscript has been judged scientifically suitable for publication and will be formally accepted for publication once it meets all outstanding technical requirements.

Kind regards,

Masoud Behzadifar

Academic Editor

PLOS ONE

Additional Editor Comments (optional):

Reviewers' comments:

Reviewer's Responses to Questions

**Comments to the Author**

1. If the authors have adequately addressed your comments raised in a previous round of review and you feel that this manuscript is now acceptable for publication, you may indicate that here to bypass the “Comments to the Author” section, enter your conflict of interest statement in the “Confidential to Editor” section, and submit your "Accept" recommendation.

Reviewer #1: All comments have been addressed

2. Is the manuscript technically sound, and do the data support the conclusions?

Reviewer #1: Yes

3. Has the statistical analysis been performed appropriately and rigorously? 

Reviewer #1: Yes

4. Have the authors made all data underlying the findings in their manuscript fully available?

Reviewer #1: Yes

5. Is the manuscript presented in an intelligible fashion and written in standard English?

Reviewer #1: Yes

6. Review Comments to the Author

Reviewer #1: (No Response)

7. PLOS authors have the option to publish the peer review history of their article (what does this mean?). If published, this will include your full peer review and any attached files.

Reviewer #1: No

---

## [Editor Report · Acceptance letter]

27 Feb 2024

PONE-D-23-24605R1 

PLOS ONE

Dear Dr. Quinn, 

I'm pleased to inform you that your manuscript has been deemed suitable for publication in PLOS ONE. Congratulations! Your manuscript is now being handed over to our production team.

Kind regards, 

on behalf of

Dr. Masoud Behzadifar 

Academic Editor

PLOS ONE